# Efficient Recovery of Solid Waste Units as Substitutes for Raw Materials in Clay Bricks

Ioannis Makrygiannis *[ID] and Athena Tsetsekou

School of Mining Engineering and Metallurgy, National Technical University of Athens, Zografou Campus, 15780 Athens, Greece
* Correspondence: ymakrigiannis@sabo.gr

**Abstract:** The advent of new materials and technologies in building materials has changed the way of building. New lighter materials with easier application methods and improved mechanical behaviors, have become necessary for the market. Moreover, the new environmental policy (2022) aims to transform the waste management into sustainable materials management to ensure the long-term protection and improvement of the environment. For the brick and tile industry, raw materials and the additives that compose the product mixture seem to be a key factor in this direction. Furthermore, every product type (solid or perforated brick) requires different additives to achieve the properties that are postulated by the international standards. For the study, the wide range of additives that were used have been assorted into three (3) categories: the inert materials, the lightweight materials, and the industrial remains. Totally, eight (8) different materials were used as additives into ceramic mass, in different proportions each time. Almost all additives used for this research were pore-forming agents. These burn out almost completely before reaching the full-fire temperature, and do not change the fired body. As a result of additives burnt out, the necessary pore volume is formed in the fired brick body, which, if combined with an appropriate percentage of voids, result in raw density readings. The pore structure is significant as long as the ultimate strength of lightweight bricks is acceptable. In this study, additives between 3 and 25% by weight were added to the clay mixture. The extrusion of specimens in solid form was carried out using the Laboratory's vacuum press. Firstly, the extrusion of specimens from the original raw material was implemented. Secondly, it was made on the material mixed with the additives mentioned above. A series of experimental activities were followed to determine the variations of the mechanical and physical properties as well as their production procedures (extrusion, drying, and firing). According to five (5) key properties measured in the current study and compared with the mixture without additives, it was found that the variation in thermal conductivity improvement is between -11% and 19%. The bending strength of the fired products showed a decrease from 16% to 55% except for the addition of bauxite residue, which increased the strength by 8%. Bigot drying sensitivity decreased from 11% to 27%. The density in two cases increased from 2% to 7% while in the majority the mixtures with the additives showed a decrease in density from 1% to 14%. Finally, the addition of the necessary water for shaping during extrusion showed a variation from a 10% decreased to a 14% increased water.

**Keywords:** solid waste; recycling materials; additives in ceramic mass

## 1. Introduction

Clay bricks are one of the oldest known building materials. A clay brick consists of clayey soil and water. They date back to 7000 BC where they were first found in the East Mediterranean area countries. The first bricks were sun dried and made from mud. Ancient Egyptians mixed clayey soil and straw and used them as main building materials. Evidence of sun-dried constructions can be seen today in ruins such as Harappa Buhen and Mohenjo-daro. As the years went by, ancient Romans started to introduce fired clay bricks in the first century of their civilization, using mobile kilns and constructing fired bricks for

the entire Roman Empire. The ancient Greeks also considered perpendicular brick walls more durable than stone walls and realized how the modern brick was less susceptible to erosion than the old-style marble walls. During the 12th century, fired bricks were reintroduced to northern Germany, creating the brick gothic period. Eventually, fired bricks became a very popular building material in varying colors and shapes in the whole Europe.

Nowadays, clay bricks are used in the construction of buildings more than any other material. There are many types of clay bricks in the market, providing different benefits depending on the needs of the constructed building. Brick selection is made according to the specific application in which the brick will be used. Standards for the brick cover specific uses of brick and classify the brick by performance characteristics. The performance criteria include strength, durability, and aesthetic requirements. The primary types of clay bricks can be assorted into two (2) different categories: solid bricks and hollow bricks. The principal difference between the two (2) types is the voids permitted (perforation). Solid bricks, according to ASTM C652-21 [1], must be a minimum of 75% solid. Hollow bricks can be classified in two (2) categories by ASTM C62-12 [2] as H40V and H60V that are distinguished by the required perforation (H40V: 25% minimum and 40% maximum perforation, H60V: 40% minimum and 60% maximum perforation). According to ASTM specification, solid bricks have a range of products, such as facing bricks, building bricks, thin bricks, paving bricks, and glazed bricks. Facing bricks are intended for use in both structural and nonstructural masonry, including veneer, where appearance is a requirement. Building bricks are intended for use in both structural and nonstructural brickwork, where appearance is not a requirement. Building brick is typically used as a backing material. Thin brick has normal face dimensions but reduced thickness. They are used in adhered veneer applications. Paving bricks are intended for use as the wearing surface on clay paving systems. As such, they are subject to pedestrian and light or heavy vehicular traffic. Glazed bricks have glaze finish fused to the brick body. The glaze can be applied before or after the firing of the brick body. These bricks may use as structural or facing components in masonry. Hollow bricks may be used as either building or facing brick but have a greater void area. Hollow bricks use fewer natural resources to manufacture. Less clay and water to form, less energy to fire, less fuel to transport—all these benefits simply by increasing voids and reducing mass, without sacrificing performance. Hollow bricks with large cell voids will permit vertical reinforcing (load bearing bricks) and grouting for structural applications, an application not possible with facing brick and present high thermal insulation for both exterior and interior walls of the building [3].

Every product type requires different additives to achieve the properties that are postulated by international standards. Breaking load, maximum absorption, drying sensitivity, thermal conductivity, and density are some of the properties that are affected by additives in the ceramic mass [4]. Additives, both natural and synthetic, act as auxiliary "raw materials" and influence many properties of fired products, such as color, mechanical strength, durability, and thermal insulation [5]. In a brick production environment, additives can be used as temper 1 to 20 wt.%. The ratio is dependent on many factors, such as availability, release of pollutants, shrinkages, porous, plasticity, and the mechanical strength of the desired final product. Several research studies have been carried out in the last decades on creating new types of bricks using additives, often consisting of residual urban and industrial materials [6–14]. In the current study, three (3) different categories of solid waste units were used for a total number of eight (8) different additives with a main scope to reduce the environmental impact such as storing these materials to storage areas. Inert materials, lightweight materials, and industrial materials remain were used (Table 1). Inert materials are wastes that do not undergo any significant physical, chemical, or biological transformations and are unlikely to adversely affect other matter with which they come into contact. Lightweight materials are the additives that are burned during the firing process of a brick and create pores on the surface of the final product. Industrial material remains are processed by-products and are characterized as industrial waste generated during factory processing.

**Table 1.** The three (3) categories and the additives materials used for the current research.

| Inert Materials | Lightweight Materials | Industrial Materials Remain |
| --- | --- | --- |
| Silica sand | Sawdust | Bauxite residues |
| Dolomite limestone | Paper sludge | Iron scraps |
| Coal | | Olive stone residues |

The tests will be focused on the construction of solid bricks and their properties. This -focusing is because of the manufacture of hollow samples internal stresses from the extruder, the mold, and also the cutting of the samples may lead to non-comparable results and the conclusions may not be reliable [15].

This work explores the possibility of using the above additives in brick production and their influence in the final product and process. The five (5) parameters with key importance, in this direction, that are measured are:

a. Thermal conductivity coefficient
b. Bending strength on the final product (fired)
c. Drying sensitivity
d. Body density
e. Added necessary water to the constructed mixture for shaping into bricks

Within this scope, the constructed samples were compared with the samples from pure clay material tested in the same preparation, extruding, drying, and firing process environment. The additives used for the study and described in Table 1, can be seen in Figure 1 and the constructed mixtures can be seen in detailed in Table 2.

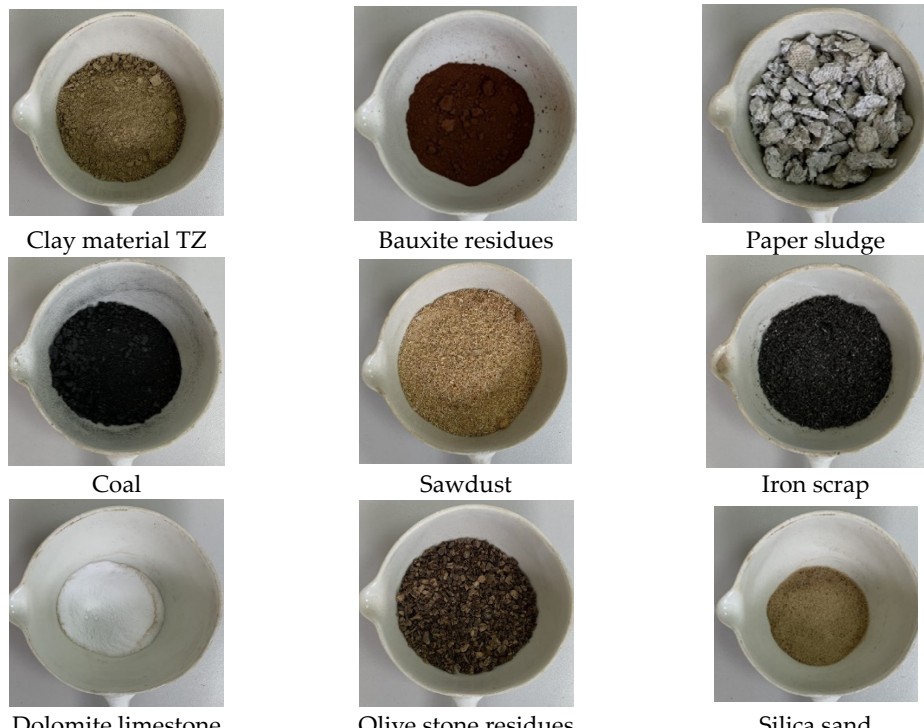

| Clay material TZ | Bauxite residues | Paper sludge |
| Coal | Sawdust | Iron scrap |
| Dolomite limestone | Olive stone residues | Silica sand |

**Figure 1.** Clay material and solid waste additives used for the study.

**Table 2.** The nine (9) labeled constructed mixtures with the ratio of each additive in the mixture.

| Mixture | TZ | TBR | TPS | TC | TSD | TDL | TIS | TOR | TS |
|---|---|---|---|---|---|---|---|---|---|
| | wt.-% | wt.-% | wt.-% | wt.-% | wt.-% | wt.-% | wt.-% | wt.-% | wt.-% |
| Clay material TZ | 100 | 97 | 92 | 92 | 96.3 | 75 | 85 | 80 | 89 |
| Bauxite Residues | - | 3 | - | - | - | - | - | - | - |
| Paper sludge | - | - | 8 | - | - | - | - | - | - |
| Coal | - | - | - | 8 | - | - | - | - | - |
| Sawdust | - | - | - | - | 3.7 | - | - | - | - |
| Dolomite limestone | - | - | - | - | - | 25 | - | - | - |
| Iron scrap | - | - | - | - | - | - | 15 | - | - |
| Olive stone residues | - | - | - | - | - | - | - | 20 | - |
| Silica sand | - | - | - | - | - | - | - | - | 11 |

## 2. Materials and Methods

### 2.1. Materials

2.1.1. Characteristics of Clay Material

The clay material used for this study are used in industrial production of Evia region. More specifically, the material is labeled TZ because it is located in an area named Triada zone, and its basic properties can be seen in Table 3. TZ can be characterized according to ISO 14688-2:2017 as an inorganic clay with medium plasticity.

The chemical composition of the clay can be seen in Table 4. The chemical analyses achieved via Atomic Absorption Spectrometry (AAS) according to ISO 26845:2016 and the suitability of the clay material can be seen in Figure 2 [16]. The grain particles of the TZ material were analyzed according to ASTM D422-63 (2007) (Table 5) and can be characterized as clayey silt material [17]. The density of the clay is on the average of 1781 Kg/m$^3$ according to ASTM D698-12 [18]. The humidity content of the clay brought from the manufacturing site to the laboratory in on the average of 8.15%.

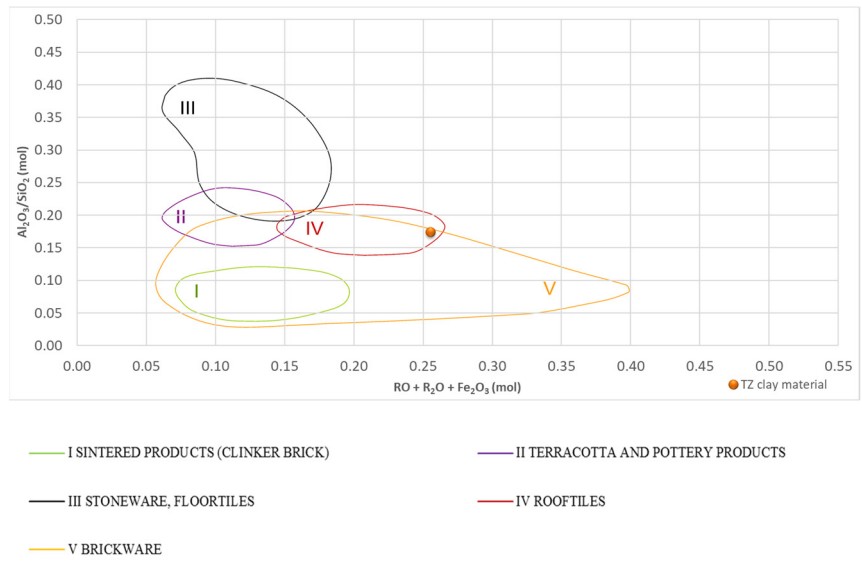

**Figure 2.** Suitability according AVGUSTINIK of TZ clay.

The XRD diffractogram of TZ clay is given in Figure 3, defining the mineralogical composition. The implementation of the XRD was achieved via Siemens D500 Powder Diffractometer. According to the XRD analysis, the TZ material consists of quartz, calcite, manganoan calcite, kaolinite, Potassium aluminum silicate, Chlorite, Palygorskite, Albite, Anorthite, and Moganite. There are number of minerals with different structural features that can be seen in a soil for brick construction [19,20].

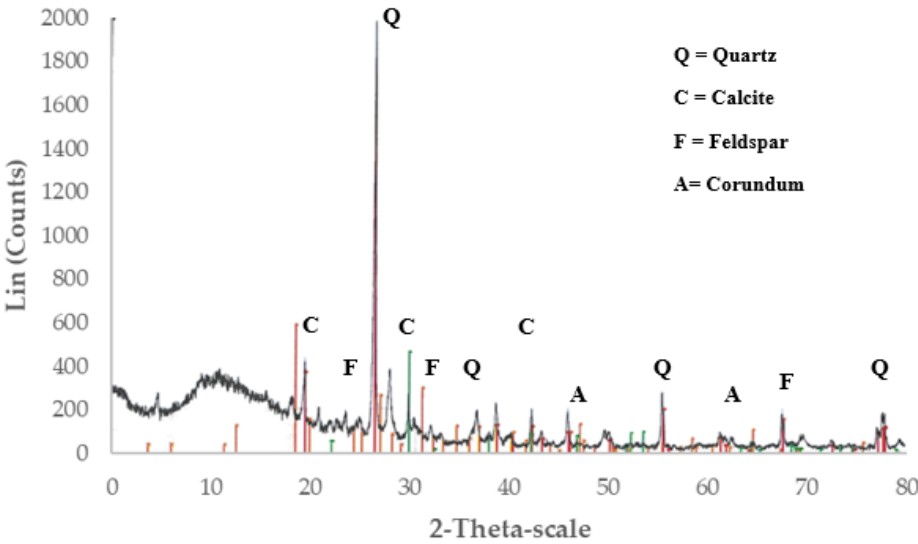

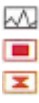

Type: 2Th/Th locked—Start: 5000◦-End: 70.000◦- Step
01-078-2315 ©—Quartz—SiO₂—Y: 86.69%—d x by: 1.- WL:1.5406—Hexagon
00-010-0173 (I)—Corundum, syn—Al₂O₃—Y: 6.17%—d x by: 1.- WL: 1.5406—R
01-083-0578 (A)—Calcite—Ca(CO₃)—Y: 22.71 %—d x by: 1.—WL: 1.5406—Rho
00-001-0349 (F)—Albite (Feldspar)—NaAlSi₃O₈—Y: 28.87%—d x by: 1.—W

**Figure 3.** XRD diffractogram of TZ clay.

**Table 3.** Physical properties of TZ clay.

| Physical Properties | Unit | Values |
|---|---|---|
| Plastic limit [21] | % | 20.76 |
| Liquid limit [21] | % | 42.00 |
| Plasticity [21] | % | 24.23 |
| Density | Kg/m$^3$ | 1781 |

**Table 4.** Oxide composition of TZ clay.

| Oxides (%) | SiO$_2$ | Al$_2$O$_3$ | CaO | Fe$_2$O$_3$ | MgO | K$_2$O | Na$_2$O | LOI |
|---|---|---|---|---|---|---|---|---|
| TZ clay | 56.45 | 16.72 | 5.65 | 5.08 | 2.73 | 1.64 | 0.55 | 9.34 |

**Table 5.** Particle size distribution of clay.

| Grain Size | Coarse Sand >63 μm | Fine Sand 63–20 μm | Silt 20 to 2 μm | Clay <2 μm |
|---|---|---|---|---|
| TZ clay | 6.00% | 9.35% | 40.37% | 44.28% |

### 2.1.2. Dolomite Limestone

In general, dolomite is a calcium magnesium carbonate having a chemical formula of $CaMg(CO_3)_2$. Limestone that contains some dolomite is known as dolomitic limestone. The dolomite limestone used for the tests is labeled DL01 and is being used as an aggregate in concrete and asphalt mixtures that are used to build highways in Israel.

The dolomite limestone presented a density of 2001 kg/m$^3$ and the grain size of its particles was less than 1 mm, as a result the use of 25% by weight, of dolomite limestone to the brick mass in order to ensure low density to the fire product without reducing the firing bending strength.

The chemical composition for the used dolomite limestone achieved via Atomic Absorption Spectrometry (AAS) according to ISO 26845:2016 and can be seen in Table 6.

**Table 6.** Oxide composition of dolomite limestone.

| Oxides (%) | $SiO_2$ | $Al_2O_3$ | CaO | $Fe_2O_3$ | MgO |
|---|---|---|---|---|---|
| DL01 | 4.47 | 0.97 | 73.97 | 0.67 | 19.92 |

### 2.1.3. Silica Sand

Silica sand, also known as industrial sand or quartz sand, is made up of two (2) main elements: oxygen and silica. Specifically, silica sand is made up of silicon dioxide ($SiO_2$). Quartz is a crystalline mineral composed of silicon dioxide—a chemically inert and relatively hard mineral. $SiO_2$ grades are at 7 out of 10 on Mohs hardness scale.

The specific type of sand was chosen and used for the current tests because is slightly less coarse and free of unwanted additives than sea sand or sandy clays. Unwanted additives in a brick and tile industry can be calcium oxide more than 18%, $K_2O$ and $Na_2O$ more than 6%, chlorine, sulfur and Fluorine more than 0.1% and organics. Pure silica sand was used as additive in a ratio of 11% in the final mixture to ensure the accepted properties of the final product (absorption capacity, mechanical strength) and the avoidance of any unwanted issue during production (quartz inversion, thermal cracks) [22].

The silica sand used for the tests was labeled ZK. The calcium carbonate was measured as zero (0) via Calcimeter Bernard Method. The particle size of the sand grains can be seen in Table 7.

**Table 7.** Particle size of the sand grains, fractions 0.063 to 2 mm.

| >2 mm | ASTM 10 | 0.00% |
|---|---|---|
| 0.71 mm | ASTM 25 | 0.15% |
| 0.60 mm | ASTM 30 | 0.15% |
| 0.50 mm | ASTM 35 | 0.90% |
| 0.40 mm | DIN 16-1171 | 4.80% |
| 0.30 mm | ASTM 50 | 19.64% |
| 0.20 mm | DIN 30-1171 | 53.82% |
| 0.10 mm | DIN 60-1171 | 19.80% |
| 0.063 mm | ASTM 230 | 0.65% |

### 2.1.4. Coal

Different types of coal were used as additives in ceramic mass for brick production, especially for lightweight products with higher thermal insulation properties. Coal is dissipated during the firing process and creates porous (small cavities in the mass) which can improve the thermal properties of the brick.

The coal used for the research is a form of amorphous carbon highly porous residues of microcrystalline graphite remains. The calorific value obtained through ASTM D-2015 using an oxygen shell calorimeter, type Parr 1341, can be seen in Table 8. The chemical analyses of the coal were determined according to iso 226845 and EN 1744-1 and can be seen in Table 9 [23]. Initially, the natural moisture of the coal was determined at a temperature of 105 °C from a laboratory oven type SCN/400/DG and can be seen in Table 8. Coal was ground down through laboratory jaw crusher model AA 92 with the opening of the jaws adjusted at 2 mm. The crushed coal was ground more using laboratory roller mill, type Verdes 080, which features two (2) smooth cylinders that work against each other at different speeds with an adjustable separation of 1.2 mm. The ground material passed through sieve with mesh 2 mm to ensure that all grain particles are less than 2 mm.

**Table 8.** Properties of coal.

| | Unit | Coal for the Research |
|---|---|---|
| Calorific value | Kcal/kg | 4120 |
| Humidity | % | 7.89 |

**Table 9.** Oxide composition of coal.

| Oxides (%) | SiO$_2$ | Al$_2$O$_3$ | CaO | Fe$_2$O$_3$ | MgO | K$_2$O | Na$_2$O | LOI |
|---|---|---|---|---|---|---|---|---|
| Coal | 1.42 | 0.26 | 0.28 | 0.12 | 0.09 | 0.14 | 0.05 | 96.50 |

### 2.1.5. Sawdust

The sawdust that was used was a local wood waste collected from a company that provides turnkey solutions and machinery for the Heavy clay sector. The pile density and the percentage of air in the sawdust used for the current research can be seen in Table 10 [24,25].

**Table 10.** Properties of sawdust.

|  | Unit | Sawdust for the Research |
|---|---|---|
| Calorific value | Kcal/kg | 4154 |
| Pile density | Kg/m$^3$ | 315 |
| Percentage of air | % | 57 |
| Humidity | % | 2.29 |

The natural moisture of the sawdust determined at a temperature of 105 °C from a laboratory oven type SCN/400/DG and can be seen in Table 10 as well. The grain size of the sawdust that was obtained was less than 2 mm. To ensure the results, the sawdust passed through 2 mm sieve (ASTM 10).

### 2.1.6. Paper Sludge

Mucahit Sutcu and Sedat Akkurt (2009) examined the use of recycled paper in processing residues for making porous brick with reduced thermal conductivity. Their conclusions gave perspective results concerning the drying sensitivity and thermal properties by reducing the density by almost 33% [26].

Carlos Maurício F. Vieira, Regina M. Pinheiro, Ruben J. Sanchez Rodrigue, Veronica S. Candido, Sergio N. Monteiro, (2016) examined the addition of 10% (wt.%) paper sludge on clay bricks and disclosed an accepted compression strength for constructions [27].

For these reasons, paper sludge was chosen to compare the final properties in clay mass with other lightweight additives in the current research.

The paper sludge used for the current research was obtained from an Israel brick and tile factory in the Beer Sheva region. The humidity of the paper sludge as obtained was 65%. The grain size could not be determined with accuracy due to the high amount of water in its mass. The sludge was mixed properly with the clay material TZ which presented 8.15% humidity. Due to this fact, the paper sludge was added with the necessary mixing water for the final mixture and was stirred steadily for twenty-four hours (24) until it was homogenized completely with the mixing water.

The chemical composition (wt.%) of paper sludge was carried out through XRF and can be seen in Table 11.

**Table 11.** Oxide composition of paper sludge.

| Oxides (%) | SiO$_2$ | Al$_2$O$_3$ | CaO | Fe$_2$O$_3$ | MgO | K$_2$O | Na$_2$O | C |
|---|---|---|---|---|---|---|---|---|
| Paper sludge | 22.3 | 12.0 | 34.6 | 0.4 | 2.1 | 0.1 | 0.3 | 22.0 |

### 2.1.7. Bauxite Residues

Bauxite residues are the solid by-product of the alumina production process. Bauxite residues are also known as red mud and it is a waste produced during the Bayer method in bauxite refining, where digestion of pulverized bauxite with sodium hydroxide at elevated temperatures and pressures takes place. It is a highly alkaline slurry with 15–30 wt.% of

solids. Red mud is composed mainly of fine particles of silica, aluminum, iron, calcium, and titanium oxides in different proportions depending on the bauxite ore, aluminum extraction conditions, and quality control [28].

The red mud used for the current research was obtained from an aluminum plant in Greece, which is one of the strongest pillars of Greek industry and has established itself as one of the strongest plants in the Metallurgy sector in the European Union. The moisture content was 18 wt.%.

The chemical analyses achieved via Atomic Absorption Spectrometry (AAS) according to ISO 26845:2016 can be seen in Table 12. The Particle-Size Analysis of the red mud was measured via hydrometer according to the ASTM D422-63 (2007). The results can be seen in Table 13.

**Table 12.** Oxide composition of bauxite residue (Red mud).

| Oxides (%) | $SiO_2$ | $Al_2O_3$ | CaO | $Fe_2O_3$ | $Na_2O$ | $TiO_2$ | LOI |
|---|---|---|---|---|---|---|---|
| Red mud | 7 | 21 | 9 | 42 | 3 | 6 | 9 |

**Table 13.** Particle size distribution of bauxite residue (Red mud).

| Grain Size | Coarse Sand >63 μm | Fine Sand 63–20 μm | Silt 20 to 2 μm | Clay <2 μm |
|---|---|---|---|---|
| Red mud | 0.00% | 0.00% | 38.61% | 61.86% |

2.1.8. Iron Scrap

Carlos Maurício F. Vieira, Lucas Fonseca Amaral and Sergio N. Monteiro (2018) examined the recycling of steelmaking plant wastes in Clay Bricks. The conclusion was that the use of 10 wt.% increases the porosity of thee fired product but decreases the mechanical strength due to the inert iron scrap on the mass [29].

The iron scrap that used for the current research was collected from a local Greek company that provides turnkey solutions and machinery for the Heavy clay sector. The iron scrap is a material free from radioactive contamination and complies with the Regulation (EC) n. 1907/2006 REACH and Directive 2011/65/EU RoHS II. The hardness, according to standard EN-ISO 6506-1 is 210 Hs. The chemical analyses of the iron scrap can be seen in Table 14. The particle size of the scrap is between 3 and 1 mm.

**Table 14.** Chemical composition of iron scrap.

| | C | Si | Fe | Mn | P | S | Ni | Cu | Cr + Mo + Ni | Al |
|---|---|---|---|---|---|---|---|---|---|---|
| | % | % | % | % | % | % | % | % | % | % |
| Iron scrap | 0.47 | 0.27 | 97.73 | 0.73 | 0.02 | 0.02 | 0.12 | 0.21 | 0.30 | 0.02 |

2.1.9. Olive Stone Residues

S. Arezki, N. Chelouah, and A. Tahakourt (2016) examined the effect of the addition of ground olive stones on the physical and mechanical properties of clay bricks. The results of the mixing of fine olive stones in a clay material gave perspective information concerning the porosity, resulting in the improvement of the thermal insulation in the final product [30].

The olive stone residues used for the research were labeled as OSR and collected from a local oil mill stood in the region of Evia Island in Greece. The organic matter of the OSR was very high and mainly composed of lignin, cellulose, fats, and hemicellulose. The chemical analyses can be seen in Table 15.

**Table 15.** Chemical analyses of OSR.

|  | Lignin (%) | Cellulose (%) | Hemicellulose (%) | Ash (%) | Fats (%) |
|---|---|---|---|---|---|
| OSR | 23.12 | 35.29 | 14.65 | 1.86 | 2.11 |

The particle size distribution of OSR was achieved with the use of sieves and the fractions can be seen in Table 16. The moisture content was 7.96 wt.% and the calorific value was measured as 4493 Kcal/kg. The OSR presented a density of 2173 kg/m$^3$ and the grain size of its particles used for the tests were less than 1 mm, as a result the use of 20% by weight, to ensure low density to the fire product without reducing the firing bending strength.

**Table 16.** Particle size distribution by sieves for OSR.

| Grain Size | Large Grains >500 μm | Medium Grains 500–63 μm | Fine Grains <63 μm |
|---|---|---|---|
| OSR | 9% | 65% | 26% |

*2.2. Methods*

The process environment, followed in this research, for the extruding, drying, and firing procedures was the same for all the construction mixtures according to industrial scale production. Firstly, the original raw material TZ was implanted and secondly the mixtures were constructed with the additives mentioned above.

2.2.1. Mixture Preparation

As the first step in every constructed mixture, the moisture content of every raw material or additive was determined. The moisture content is the share of water existing in a humid raw material or in a humid additive, given in percent, in which the quantity of water refers to the weight of the humid material (wet base). The sample is weighted into a laboratory vessel for the determination of the moisture. This sample is dried in the laboratory electric dryer type SCN/400/DG, at 105 °C for twenty-four (24) hours. A constant weight of sample is reached, and it is weighted again. Then the weight of the laboratory's vessel is subtracted from the total weight.

The materials were pre-crushed with a jaw hammer (model A92) with an opening of the jaws at 2 mm. Then, the crushed materials pass through a laboratory roller mill, type Verdes 080 with an adjustable separation between the two (2) cylinders of 1.2 mm. The paper sludge, as mentioned above, was mixed with the necessary extruding water and stirred for twenty-four (24) hours. The materials were weighted according to their mixing ratio, considering their actual moisture, and then were homogenized according to the recipe of every mixture separated in a kneading mixer where the necessary preparation water added. The addition of water was continued until a satisfactory plasticity index (Pfefferkorn's test) [31]. The Pfefferkorn plasticity method is based on the verification of the deformation of the sample as a result of the fall of the calibrated plate on the underlying test body shaped by means of the ancillary shaping tool. The Pfefferkorn test has two (2) reading scales: one measures the deformation in mm; the second one determines the test body deformation according to the Pfefferkorn theory. For the current study, the Plefferkorn plasticity tester that used was the Ceramic Instruments 01CI4540 and the calculation method was described by Amorós et al. [32]. The water addition is always unique in the pre mixture and depends on the absorptivity of the clay material and the extrusion process that will be followed for a given type and final product.

### 2.2.2. Extrusion

The homogenized body of every mixture was extruded through vacuum-extruded rectangular samples of standard dimensions for all tested mixtures. The laboratory extruder is model HANDLE KHS-Type: PZVM8b. The wet material after mixing was placed to the feeding chamber on top of which there is a porch for material input, following a pre-extruder mixer which includes a screw mixer that pushes the material through an-air vacuum chamber to the extruder's output. The pressure was monitored through a pressure gauge. The outer extruding part can receive interchangeable molds in the desired size and shape of the extruded products. All extruded samples were solid (no hollows on their mass) with a size of 120 × 20 × 20 mm (L × W × H). The vacuum pressure was stable and the same for all tested mixtures and equals to 0.8 $k_p$/$cm^2$ (Figure 4).

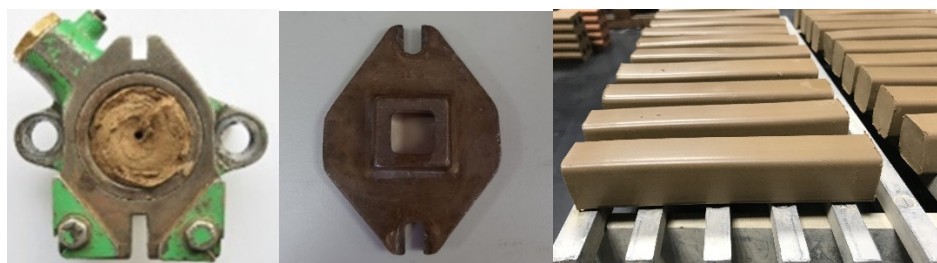

**Figure 4.** The vacuumed mixture in the extruder, the sample mouth, and the constructed pieces.

The plasticity, according Pfefferkorn for all mixtures, was between 0.7 and 0.9 by adding the necessary water. Fifteen (15) samples were constructed for every test and in total, for all nine (9) mixtures, 135 samples (9 mixtures × 15 samples).

### 2.2.3. Drying

All the extruded specimens were marked and placed in the laboratory electric oven (type SCN/400/DG) for a smooth drying circle [33]. The drying circle consisted of three (3) different phases. The humidity phase, in which the humidity in the dryer was kept at 90%. The "shrinkage" phase, in which the drying rate drops dramatically, and the shrinkage of samples approaches the end. And the "drying" phase in which the ambient humidity is almost zero, and the temperature is at the maximum to secure the loss of the remained humidity in the samples. The drying circle and the phases followed for the smooth drying in this research can be seen in Figure 5.

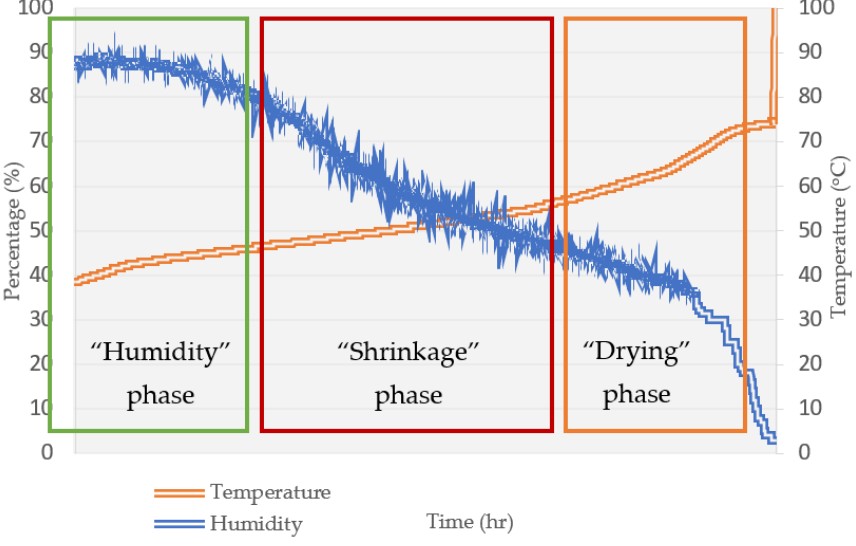

**Figure 5.** Drying circle and each of three (3) phases followed for the tests.

Each phase demands specific attention because each phase could create different issues on the samples. During "humidity" phase, ambient humidity in the dryer must be in high levels to keep the surface pores of the bricks open. It is the most critical phase during drying because cracks, deformations, or fragility of the bricks may occur. During the "shrinkage critical point phase" the drying shrinkage should be complete before the temperature rises rapidly to complete drying. In this phase, the temperature rise should happen gradually to avoid cracking issues. In the last phase, the target is to diminish as much as possible the remaining body humidity in the bricks. All regulations should follow this demand and adjust to the production mixture and its behavior.

At the beginning, the main scope of the drying is the surface pores of the samples to be kept opened to facilitate the loss of humidity from the internal body. This face is the most critical because in the second phase during which the temperature rises and the humidity of the dryer drops, cracks, deformations, or fragility may occur. Dried samples of the study can be seen in Figure 6.

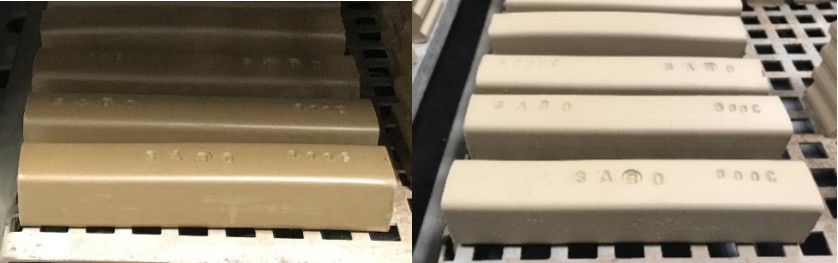

**Figure 6.** Samples placed in the laboratory dryer before and after the drying procedure.

Bigot's Curve

Drying sensitivity of the constructed mixtures was determined through Bigot's curve for every mixture separately [34]. The tested samples through Bigot's curve were placed in the laboratory's dryer for a tested temperature of 25 °C and stable moisture inside the dryer of 75%. This curve is a graphic representation of water content's variation as a function of linear shrinkage percent measured for twenty-four (24) hours' time. According to Bigot's curve method, the drying curve consists of two parts. The first part is linear with a very high linear correlation I which indicates the constancy in the drying rate. The linear part thus the constant drying rate stops when the critical point is reached. In the second part, drying continues at a decreasing rate and the shrinkage gradually approaches the end.

The laboratory dryer was equipped with all the necessary instruments to study the green brick samples' drying shrinkage behavior. The dryer was composed, as described above, of three major sections: the tunnel dryer unit, the air preparation unit, and the control system (measurement sensors and data acquisition). The tunnel dryer had approximately a volume of 125 cm$^3$. It was insulated to prevent heat loss to the surroundings. An adjustable centrifugal fan and an adjustable electrical heater were placed in the air preparation unit. The drying air was supplied by the centrifugal fan from the ambient. The power and the capacity of the fan were 32 W and of 85 m$^3$/h, respectively. Temperature of the drying air was regulated by a PID-controlled (Jumo Dtron 304). Ambient air initially passed through an electrical heating zone and later flowed over the sample. The heated air was moved through the tunnel dryer where the sample holder was located 35 cm away from the air inlet. Thus, air flows parallel to the surface of the sample which is located on the wire mesh. The distance between the hot air inlet and outlet was 75 cm in the tunnel dryer unit. Initially, the air fan and electrical heater were utilized to obtain the steady-state test conditions. Then, a green brick sample was placed on the metallic carrier. The drying air humidity was observed throughout the experiments. The relative humidity of the air inside the dryer was measured every 5 min by a humidity sensor (TMI Orion—CeriDry). The exact measurements and methodology described by Mancuhan, Sargut, and Ozit [35].

The sensitivity level is calculated by the index CSB, which is the indicator of sensitivity in drying according to Bigot and is classified referring to the Table 17.

**Table 17.** Classification of drying sensitivity according to Bigot's CSB index.

| Classification of CSB | |
| --- | --- |
| <1.0 | Insensitive |
| 1.0–1.5 | Medium sensitive |
| 1.5–2.0 | Sensitive |
| >2.0 | Highly sensitive |

### 2.2.4. Firing

The firing procedure was carried out in a laboratory's electrical gradient kiln type Nabertherm model GR1300/13. The kiln is computer controlled with ten (10) time-duration steps to ensure the adequate preheating, firing in max temperature and cooling of the samples. All samples stayed at the maximum peak temperature of 900 °C (average temperature for a brick and tile industry for building bricks) for three (3) hours. The preheating and cooling phases, especially close to 573 °C, were smooth to avoid any issues from quartz inversion [36].

The rate of increasing the temperature was between 0.7 and 1.16 °C/min, depending on the firing zone and the programmed procedure from cold to cold was twenty-four (24) hours. The firing circle can be seen in Figure 7.

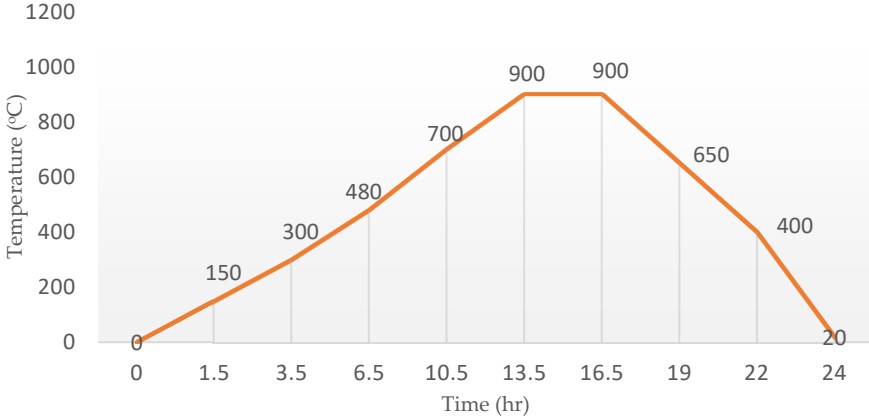

**Figure 7.** The firing circle followed for every tested mixture.

All the mixtures presented no issues during firing. The samples were totally dried before entering the kiln as they had been placed in the dryer for twenty-four (24) hours at 105 °C.

### 2.2.5. Measurements

The weight losses were determined using the laboratory scale type Kern FKB 36K0.2 The linear drying shrinkage and firing linear shrinkage were measured in accordance with the standard ASTM C326-09 [37].

The Archimedes method based on ASTM C373-14a [38] was used to determine the bulk density and water absorption. After absorption, efflorescence tests were carried out to evaluate the alkaline salts present in the bricks according to ASTM C67-21 [39]. The thermal conductivity (λ 10, dry, mat) of the fired clay samples was calculated according to the EN1745:2012 [40].

The bending strength of the fired samples was measured in test specimens with dimensions of 120 × 20 × 20 mm. A three-point bending test device with a distance of 100 mm between the bearings was used for this purpose and 3 test specimens from each composition/production method were tested.

To calculate the water content after drying, 15 test samples from every preparation procedure, with dimensions of 120 × 20 × 20 mm, were weighted directly after shaping and dried in the laboratory's oven using a twenty-four (24) hour drying cycle as described above. The preparation water content was calculated from the wet and dry weight according to the following formula:

$$\text{WR} = \frac{Weight\ of\ wet - Weight\ of\ dry}{Weight\ of\ dry} \times 100 \tag{1}$$

In order to avoid any misunderstandings, the weight of the dry specimens was used as a reference throughout.

## 3. Results

The results came out from the constructed mixtures, of which the same number of constructed samples (fifteen) were tested through the same process environment.

Nine (9) mixtures constructed with the ground TZ clay and the additives. The additives were added to the TZ clay in a ratio between 0 to 25 wt.% depends on the density, granulometry and calorific value of every additive. The mixing of clay with the additives (except paper sludge) was achieved through a conical rotated mixer model MI/10, for forty (40) minutes.

A series of experimental tests (preparation, extrusion, drying and firing) were followed to determine the variations of the mechanical and physical properties as well as their production environment. The experimental tests for every constructed mixture labeled with abbreviations to discriminate the samples. TZ clay with bauxite residues labeled TBR, TZ clay with dolomite limestone labeled TDL, etc. Significant focus was given on three (3) basic sections of the production process especially in five (5) key-parameters that can be seen in Table 18.

**Table 18.** Variations of mechanical and physical properties on three (3) sections.

| Extrusion | Drying | Firing |
|-----------|--------|--------|
| Extrusion water | Sensitivity | Thermal conductivity |
| Plasticity | Shrinkage | Body density |
| | Bending Strength | Bending Strength |

### 3.1. Extrusion Results

Based on the extrusion procedure, the mixture with 8% paper sludge presented a significantly increased plasticity of the wet mixture in almost the same amount of extruding water with the 100% clay mixture. Sawdust, olive stone residues, dolomite limestone, and coal additions presented better plasticity. However, the demands of the necessary extrusion water also increased significantly. Silica sand, iron scraps and red mud (bauxite residues) presented decreased plasticity in comparison with the 100% clay, resulting decreased water demands for the extruding. The plasticity and the necessary water for all mixtures can be seen in detailed in Figure 8.

### 3.2. Drying Results

Based on the drying results, the sensitivity is decreased for all additives added to the clay material in comparison with the 100% clay. Significant reduction in sensitivity showed the mixture with the paper sludge. Regarding the shrinkage values, the iron scrap mixture presented the lowest in contrast with the dolomite limestone mixture which presented the highest. Concerning the bending strength of the dried samples, the olive stone residues mixture showed the highest values followed by the bauxite residues mixture. The lowest bending strength results presented by silica sand and coal mixtures, respectively. The results carried out from the drying procedure can be seen in detailed in Figure 9 for shrinkage and Bending strength and Figure 10 for sensitivity.

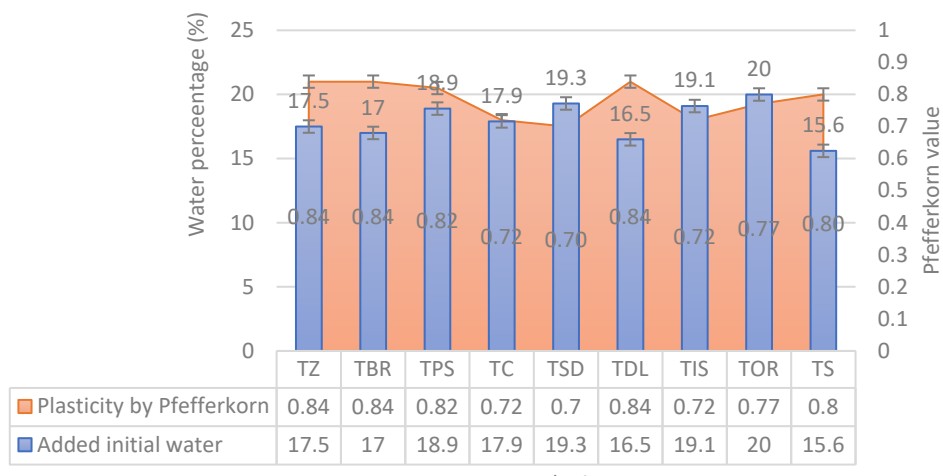

**Figure 8.** Plasticity by Pfefferkorn and necessary extrusion water for all mixtures.

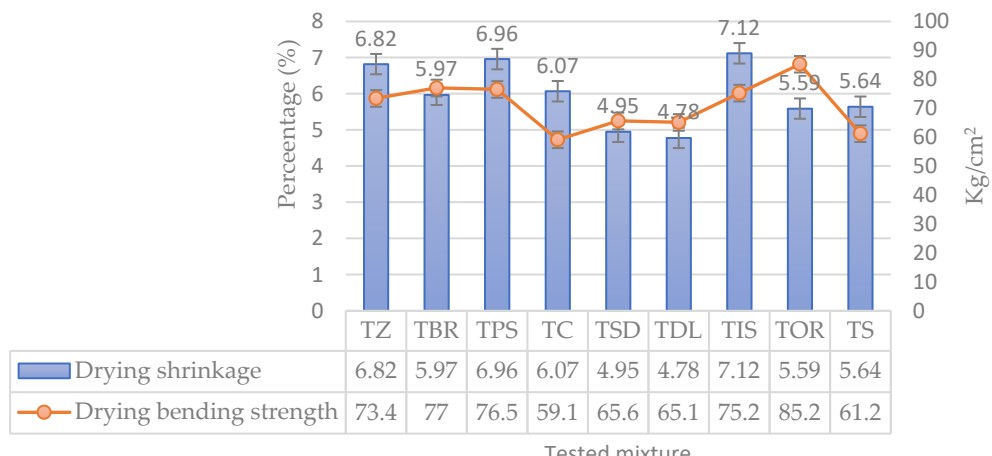

**Figure 9.** Drying linear shrinkage and Bending strength of the dried samples.

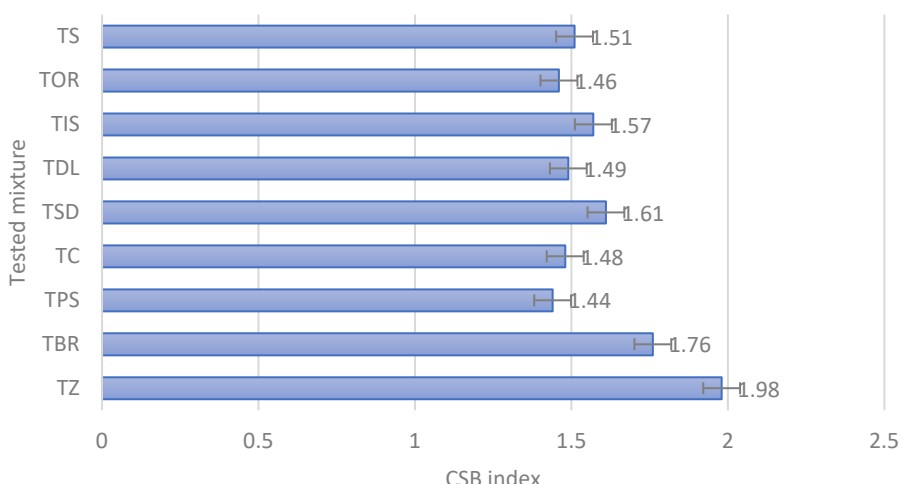

**Figure 10.** Drying sensitivity according Bigot's curve results (CSB).

*3.3. Firing Results*

The firing results carried out after the implementation of the firing circle and the evaluation of the samples for any deformations, cracks or issues that affect the accuracy of the results. According to the EN1745:2012, the thermal conductivity of the fired clay

samples is related to the density of the fired body (36). According to EN1745:2012, the thermal conductivity of the samples from 100% clay material presented a thermal conductivity (λ value) equal to 0.52 W/m·K. Most tested additions showed improved thermal conductivity in comparison with 100% clay. The lowest thermal conductivity was shown by sawdust, paper sludge and coal additions. Dolomite limestone and olive stones residues presented also have improved lambda values. Silica sand addition presented almost the same results with minor improvement. Only bauxite residues and iron scraps additions presented the worst thermal conductivity behavior. The experimental thermal conductivity results can be seen in detailed in Figure 11.

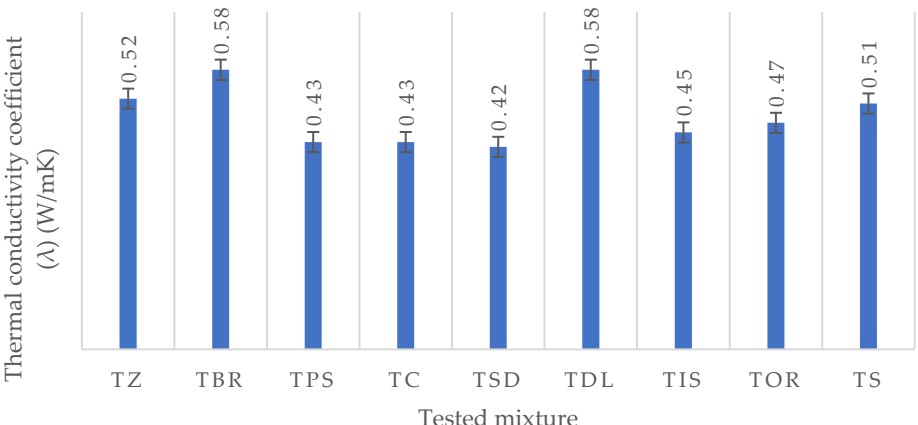

**Figure 11.** Thermal conductivity values for every tested addition including 100% clay.

The changes in the bending strength values in the tested mixtures for all constructed mixtures can be seen in Figure 12. All lightweight additions except Bauxite residues (red mud) that were used for the mixtures presented significant reduction of the bending strength, compared to the 100% clay samples. Bauxite residues showed increased bending strength due to the aluminum oxide that contain in their composition. However, all the mixtures fired at 900 °C showed strength values higher than 100 Kg/cm$^2$.

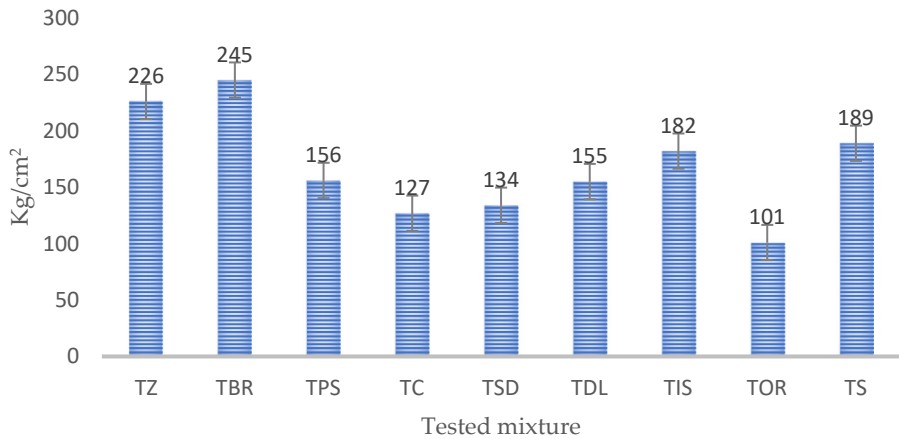

**Figure 12.** Bending strength values of every tested mixture (900 °C).

Water absorption is important for a construction clay brick because the plaster is applied on their surface. Based on the American Standard of Testing Material (ASTM, C67-94) the percentage of water absorption should not be more than 18% by weight. Lowest values lead to face bricks that plaster is not necessary. Iron scraps was the only tested additive that presented lower absorption capacity than the 100% clay mixture. The highest absorption capacity showed by olive stone residues. Sawdust and paper sludge addition

also presented a significant increase in absorption capacity. In detailed the absorption values results for all mixtures can be seen in Figure 13.

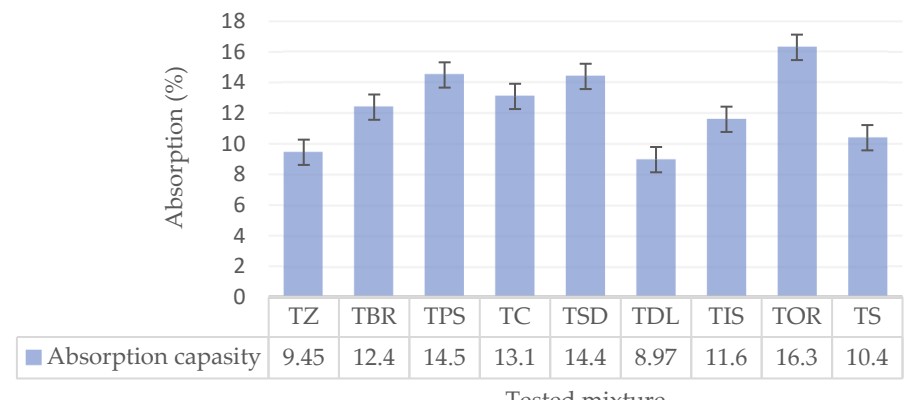

**Figure 13.** Absorption capacity value for the fired tested mixtures.

To valorize the use of the tested additives in the clay ceramic mass, the results derived from the process environment of the research were gathered and can be seen in detail in Table 19.

**Table 19.** Mixture proportions and gathered results.

| Mixture | TZ | TBR | TPS | TC | TSD | TDL | TIS | TOR | TS |
|---|---|---|---|---|---|---|---|---|---|
| Plasticity by Pfefferkorn | 0.84 | 0.84 | 0.82 | 0.72 | 0.70 | 0.72 | 0.84 | 0.77 | 0.80 |
| Added initial water (%) | 17.5 | 17.0 | 18.9 | 17.9 | 19.3 | 19.1 | 16.5 | 20.0 | 15.6 |
| Linear dry shrinkage (%) | 6.82 | 5.97 | 6.96 | 6.07 | 4.95 | 7.12 | 4.78 | 5.59 | 5.64 |
| Bending strength (dry) (Kg/cm$^2$) | 73.4 | 77.0 | 76.5 | 59.1 | 65.6 | 75.2 | 65.1 | 85.2 | 61.2 |
| Bigot's curve (CSB) | 1.98 | 1.76 | 1.44 | 1.48 | 1.61 | 1.57 | 1.49 | 1.46 | 1.51 |
| Firing temperature (°C) | 900 | 900 | 900 | 900 | 900 | 900 | 900 | 900 | 900 |
| Weight loss dry-fired (%) | 8.2 | 7.2 | 12.7 | 15.5 | 10.7 | 16.8 | 2.7 | 16.3 | 7.7 |
| Weight loss wet-fired (%) | 25.7 | 24.3 | 31.6 | 33.5 | 30.0 | 36.0 | 19.2 | 36.4 | 23.4 |
| Firing shrinkage (%) | 0.05 | 0.32 | 0.01 | 0.31 | 0.23 | 0.04 | 0.05 | 0.04 | 0.00 |
| Linear total shrinkage (%) | 6.87 | 6.29 | 6.97 | 6.38 | 5.18 | 7.16 | 4.83 | 5.63 | 5.64 |
| Water absorption (%) | 9.45 | 12.4 | 14.5 | 13.1 | 14.4 | 11.6 | 8.97 | 16.3 | 10.4 |
| Bending strength fired (Kg/cm$^2$) | 226 | 245 | 156 | 127 | 134 | 182 | 155 | 101 | 189 |
| Body density (Kg/cm$^3$) | 1.72 | 1.76 | 1.49 | 1.49 | 1.48 | 1.55 | 1.85 | 1.60 | 1.70 |
| Thermal cond. Coefficient (W/m·K) | 0.52 | 0.58 | 0.43 | 0.43 | 0.42 | 0.45 | 0.58 | 0.47 | 0.51 |

## 4. Discussion

Within the scope of this research, eight (8) different additives were added to a clay material in different proportions and totally overall nine (9) different mixtures were constructed. All additives were solid waste materials assorted in three (3) categories: inert materials, lightweight materials, and industrial remains.

The main target was to determine the variations of the mechanical and physical properties (Table 20).

The experimental results obtained during the mixing and extruding tests for all three (3) categories of additive materials used showed variations in the mixing water, which changes the plasticity of the mixture and affects the drying process. Specifically, in Table 20, the lightweight materials presented higher water needs for their formation. Compared with the 100% clay material, an increase of 9% of the necessary water was determined. Dolomite limestone and olive stone residues also showed a large increase in water, while all the other additives did not present significant variation in the amount of admixture water. This fact is coming in accordance with the theory that these materials tend to bind

water on their surface, which during extrusion passes to the clay molecules, increasing the plasticity [41]. The mixtures that presented a lower percentage of necessary mixing water are because these materials do not absorb water, such as bauxite residues, silica sand, or iron scraps with the result of reducing the plasticity of the mixture.

**Table 20.** Summary of results from the constructed mixtures.

| Mixture | | TZ | TBR | TPS | TC | TSD | TDL | TIS | TOR | TS |
|---|---|---|---|---|---|---|---|---|---|---|
| | | wt.% | wt.% | wt.% | wt.% | wt.% | wt.% | wt.% | wt.% | wt.% |
| Laboratory code: | | 100% clay material | 3% bauxite residues | 8% paper sludge | 8% coal | 3.7% sawdust | 15% iron scrap | 25% dolomite limestone | 20% olive stone residues | 11% silica sand |
| Peak firing temperature | °C | | | | | 900 | | | | |
| Bending strength/fired | Mpa | 226 | 245 | 156 | 127 | 134 | 182 | 155 | 101 | 189 |
| Body density | Kg/cm$^3$ | 1.72 | 1.76 | 1.49 | 1.49 | 1.48 | 1.55 | 1.85 | 1.60 | 1.70 |
| Drying sensitivity | CSB | 1.98 | 1.76 | 1.44 | 1.48 | 1.61 | 1.57 | 1.49 | 1.46 | 1.51 |
| Added water (for shaping) | % | 17.5 | 17.0 | 18.9 | 17.9 | 19.3 | 19.1 | 16.5 | 220.0 | 15.6 |
| Thermal conductivity ($\lambda_{eff}$) | W/m·K | 0.52 | 0.58 | 0.43 | 0.43 | 0.42 | 0.45 | 0.58 | 0.47 | 0.51 |

Based on this study's experimental firing investigations, it has been determined that lightweight additives are sublimated during the firing process and create porous. Porous are small air cavities in the body of the clay brick sample decreasing the density of the final product. This effect is decreasing the lambda value of the fired clay mass ($\lambda_{eff}$) increasing the thermal insulation of the final product.

Another advantage of using lightweight additives in ceramic mass is that the additives are giving heat-producing energy during the preheating area depending on the calorific value of the additive, which affects the overall energy consumption of the firing procedure [42,43]. However, the final strength of the product is decreased due to the porous on the ceramic body.

The inert materials additions in the clay mass presented mediocre results concerning the final products. However, they are necessary to improve the process environment. The addition of such materials presented decreased drying sensitivity, which leads to save drying time during production. Also, it increased slightly the porosity of the final product, resulting in a slight increase in the thermal insulation properties and the absorption capacity.

The added proportion needs always to be evaluated because a high ratio of these types of additives can create chain issues on the quality of the products. High ratios are decreasing the plasticity of the wet material, making it difficult to be extruded as high perforated brick. Also, if these materials are concerning organic mass such as coal, they can create issues during firing process such as an unstable firing zone [44]. Depending on the high calorific value of the additive, the temperature in the kiln is increasing faster than programmed.

Another issue observed from calorific value rich materials as additives is the oxygenation of their bodies inside the kiln. If there is not enough oxygen during firing, color differences were observed on the surface of the samples.

## 5. Conclusions

The industrial remains that tested for the current study presented different results concerning the process environment and the final product. Based on the current study's numerical investigations, the additives are affecting the final product made by 100% clay material in different ways.

The addition of 3% bauxite residues is giving an increase of body density by 2.3% which is leading to an increase in thermal conductivity coefficient ($\lambda_{eff}$) by 11.5%. The bending strength is improved by 8.4%. Drying sensitivity is decreased by 11% and the necessary water for shaping was decreased by 2.9%. The color of the final product changed significantly to darker reddish at the tested peak temperature of 900 °C.

The addition of 8% paper sludge led to a decreased index of body density by 13.4% and as a result of this fact, the thermal insulation coefficient was reduced by 17.3%. The bending strength of the fired product was decreased by 31% while the necessary water for the extrusion was increased by 8%. The drying sensitivity, however, was decreased by 27.3%. By adding a coal ash in a ratio of 8% by weight, the density and thermal insulation coefficient were reduced by 13% and 17%, respectively. The bending strength is reduced significantly by 44% while the necessary water was almost constant as it increased slightly by 2%. Drying sensitivity was also decreased by 25%. However, a significant issue when adding coal ash was the color differentiation of the fired products resulting from the anaerobic combustion between coal and oxygen inside the kiln. Another issue was resulted from the coal ash when increasing the temperature above 700 °C as an exothermic reaction occurs, which increases the temperature faster than the adjustable program of the kiln, which makes the firing zone difficult to be controlled. The proportion of 3.7% sawdust in the clay material presented a density reduction of 14% which led to an improvement of thermal conductivity coefficient by 19%. The bending strength was decreased by 40% and the necessary water for extrusion increased by 10%. The drying sensitivity decreased by 18%.

The olive stone residues showed similar results with sawdust addition, with better plasticity and drying behavior. More specifically, by adding 20% olive stone residues, the density is reduced only by 7% and the thermal insulation improvement was 8.2%. The bending strength decreased by 55% while the necessary water for extrusion increased by 14%. The drying sensitivity, however, decreased by 26%. The iron scrap presented higher density (by 7.5%) with decreased final strength comparing with the 100% clay (by 31%). Another significant point observed when using iron scrap as an additive is that the iron pieces remained in the ceramic body after firing, making the appearance not attractive and the bonding of the sample not so strong. The thermal insulation was increased by 11% in contrast with the reduced necessity of the water for extruding (decreased by 6%). Dolomite limestone in ratio of 25% led to minor changes in the properties of the final product. The density reduced by 9.8% while the improvement of thermal insulation coefficient was 13%. The bending strength was reduced by 19% while the water needs for the extruding increased by 9%. Like all additives, the drying sensitivity reduced by 20%. Silica sand addition of 11% in the ceramic mass is a standard and easy way for a brick and tile industry to reduce the drying sensitivity and to improve the process environment of a brick factory. In our research, the results presented a minor improvement in thermal insulation coefficient by 2% and a decreased density value by 1%. The bending strength was reduced by 16%, the necessary water for shaping reduced by 11% and the drying sensitivity decreased significantly by 24%.

Based on these preliminary results, it can be concluded that the additives are helping both the process environment and the final product. The surface appearance and the color differences can be seen in Figure 14.

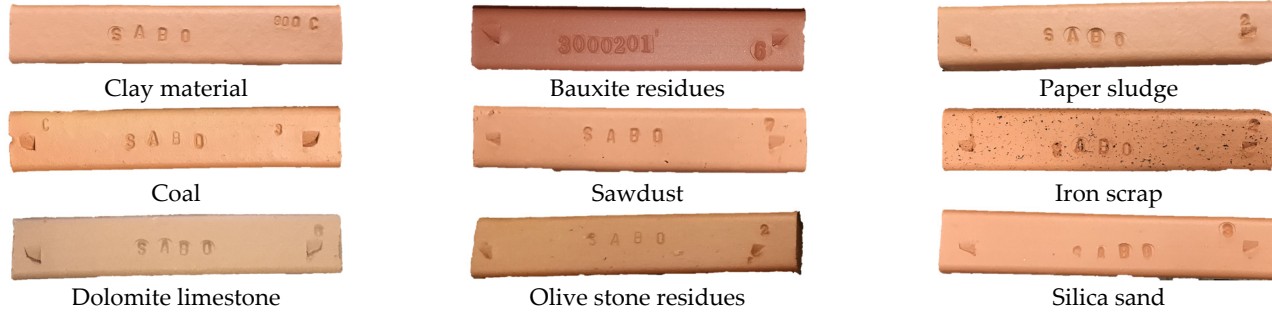

**Figure 14.** Surface color and appearance of the sample bodies constructed from the tested mixtures.

With the above results, we can see that any additive that did not contain $Al_2O_3$ reduced the drying sensitivity, a fact very important to increase the productivity of a

ceramic factory. In addition, the combustion, especially of organic additives (inert and lightweight materials), creates porosity in the ceramic mass. The porosity results in the reduction of density, which is linked to the improvement of the thermal conductivity of the final product, which is an utmost need for today's market.

The addition of additive materials to the ceramic mass, however, in a ceramic factory, should start from very small proportions (less than 3%), increasing the ratio gradually since the necessary water for extrusion is significantly reduced by using additives and raises the pressure of the extruder. The strength of the final product is also significantly reduced due to a. the creation of porosity in the ceramic mass (i.e., air cavities in the material mass) and b. the decrease in $Al_2O_3$ percentage as $Al_2O_3$ makes the cohesion of the material particles stronger [45].

A future task within the scope of the research project "recovery of solid waste units in brick mass" will concentrate on the following main target: the use of refuse derived fuel (RDF) in the ceramic mass for building clay bricks.

**Author Contributions:** Conceptualization, I.M. and A.T.; methodology, I.M.; validation, I.M. and A.T.; formal analysis, I.M.; investigation, I.M.; resources, I.M.; data curation, I.M. and A.T.; writing—original draft preparation, I.M.; writing—review and editing, I.M. and A.T.; supervision, A.T. All authors have read and agreed to the published version of the manuscript.

**Funding:** "The APC was funded by 1400 CHF by "COMPETITIVENESS, ENTREPRENEURSHIP, AND INNOVATION" (EPAvEK)" through NATIONAL ACTION: "RESEARCH-CREATE-INNOVATE SECOND CYCLE" as part of the project with Acronym RES.U.REC.T with work code T2EΔK-03668.

**Data Availability Statement:** The data presented in this study are available on request from the corresponding author.

**Acknowledgments:** This study was performed as a part of RES.U.REC.T Project no T2EΔK-03668. The authors are grateful to SABO S.A. staff for providing details for a brick and tile industry operation and in particular the clay laboratory department, for providing the extruding equipment and the support on the experimental process environment of the study. Special acknowledges to the XALKIS S.A. company for providing the clay material with the code TZ.

**Conflicts of Interest:** The authors declare no conflict of interest.

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
