# Peer review of "Efficient Recovery of Solid Waste Units as Substitutes for Raw Materials in Clay Bricks"

_recycling, doi:10.3390/recycling7050075_

Round 1

Reviewer 1 Report

The authors attempted to show different mixtures of waste materials in clay. The manuscript presents a rare example of using the same methodology and the same clay to try various mixes in fired bricks. However, the text and results need to be improved to be of better clarity and brevity, as follows.

-          Please, add information on the waste materials attempted and the mixing quantities in the abstract section.

-          The structure of the manuscript should be changed. Chapter 4. should be placed before any other results (to be chapter 3). Chapter 2. should contain Materials and methods, but not the results. Discussion should be the last chapter before the Conclusions section. Table 1 does not belong to the Introduction section, but Materials and methods.

-          More relevant publications are required to compare your results to those that used these materials (coal ash, sawdust and paper sludge).

-          Is coal or coal ash used? Improve the information in Tables 2 and 11 and the text.

-          It should be better to use some logical abbreviations to discriminate the samples, for example, TZ+bauxite residues=TBR or TZ+sawdust=TSD, etc.

-          Did you follow efflorescence after absorption tests? Why were so high contents of dolomite limestone and olive stone residues selected?

-          How were Bigot and Pfefferkorn tests conducted and in which instruments? Refer to previous publications. How were the results obtained and calculated? The results should be compared to the literature. The result concerning the mixture TC7 is strange, given the quantity of water. These parameters should follow each other: mixing water changes the plasticity and drying susceptibility, as well as drying shrinkage. Please, discuss them all together. Explain in detail the reasons for increasing or decreasing these parameters when waste is added.

-          Explain all the results and compare them to the literature. Why is thermal conductivity (here wrongly called „thermal coefficient“) changed when different wastes are used?

-          Based on which reference is the claim the bricks are acceptable to absorb between 12 and 18 % of water? There are different classes of bricks considering these different ranges, in various standards.

-          Table 4 is unnecessary since it repeats already presented information.

-          The discussion section does not contain any reference to previous literature.

-          Please, quantify the results presented in Table 5 and show the percentage of the change introduced by waste.

-          The interpretation of the XRD results is wrong. Which clay minerals are present, are there any feldspars?

-          The chemical analysis of the iron scrap does not seem good. What is contained in the rest of the material?

-          The conclusion section should be without a table.

-          Please check English and grammar and make sure every sentence contains the subject and predicate and finishes with the dot.

-          Please, change „drying bending strength“ to „dry bending strength“.

Reviewer 2 Report

This manuscript aims to transform waste management into sustainable materials management, to ensure the protection and improvement of the environment and public health. This work explores the possibility of using the above additives in brick production. As such, it will bring insightful and valuable knowledge to the world in managing waste materials and could be helpful as new building materials. This is one solution that is highly recommended to cater for this issue. Some corrections in this manuscript should be made accordingly.

COMMENTS
1. Line 12. Regarding to cater public health issue, what is the methods that you used in your research? What is public health issue that come from improper waste management?
2. Line 22. What do you mean by production environment?
3. Line 33-89. Lack of citation in the Introduction part. Authors should acknowledge previous researcher accordingly.
4. Line 52. Introduction. As you mentioned, primary types of clay bricks can be assorted in two (2) different categories, solid bricks, and hollow bricks. What are your research is focusing and why?
5. Almost all paragraphs in introduction are about clay bricks. But we do not know enough about the other 8 materials like the origin and sources. Please revised.
6. Line 99 – 105. It should be on subtopic Materials and methods. The structure of this manuscript should be revised and rearranged. Supposedly subtopic for materials and methods are after the Introduction not after the results.
7. The size of particles for each material. Please put in one table to compare the suitability of each sample. It is one of the key factors in understanding their mechanical and physical properties example compressive strength and so on.
8. Line 111. What is TZ meaning? Authors should clarify what is TZ first before using shortform.
9. Line 122. Paper sludge (TC2) increased plasticity of the wet mixture? Why. How about dolomite stone also look similar value?
10. Line 168. Add some critical discussion on the differences of thermal coefficient values. Why?
11. Line 177. What is the reason TC8 shows the highest absorption capacity?
12. Line 183. Table 4. Please remove the mixture proportion because it repetitive.
13. Line 219. Materials and methods first before results….
14. Line 233. XRD analysis after added additives??
15. Line 281. Please recheck numbering of tables in this manuscript and suggest putting the oxide composition of all the additives in one table.
16. Line 402. All extruded samples were solid (no hollows on their mass) with a size of 120 x 20 x20 mm (L x W x H). Is this an international standard size for clay brick? Why you choose this size? I understood that it is used for extrusion but is this size is comparable to standard size of brick?
17. Line 487. What is the indicator of the final product color? What can you evaluate from the color?
18. Line 546-548. Basically, we don’t put summary of results in the conclusion part. Please revised.
19. Please recheck the format of the references and suggest to put latest references.

Reviewer 3 Report

Your manuscript has been preliminary evaluated. The paper focus an interesting issue. My comments can be found below:

1.  Section; 4 Materials and methods: Author should determine the XRD composition of all raw materials used in experimental.

2.  Section; 2 Results: Authors need to add the significance section of this study with reference to past studies for highlighting the novelty of this study. What is the novelty of this work? Please discuss with reference to similar past studies or reports.

3.  Section; 2 Results: How many times for measurement.? Please provide and present the standard deviation.

4.  Section; 4.2.4. Firing: The scope of the work is rather narrow (e.g., focusing on 900 C temperature), and it is not clearly described in the text why the scope and directions of the study were taken.

5.  Author should add the results about SEM image or light microscope of samples, which are added different solid waste after firing process. (Results and discussions)

Round 2

Reviewer 1 Report

The manuscript has been improved. However, due to serious mistakes concerning clay minerals identification, I think it must be improved again before the publication.

1.      Mineralogical analysis is not right as written on the graph obtained. How did you record nitromagnezite? “Quartz and calcite are the main clay minerals that are present to the TZ clay xrd and now it is coming in accordance with the chemical analyses results.” Quartz and calcite are not clay minerals!!! Please, consult more literature on these materials… Feldspars are also always present in these materials…

2.      Besides, as it is written on the journal`s website: “We do not have strict formatting requirements, but all manuscripts must contain the required sections: Author Information, Abstract, Keywords, Introduction, Materials & Methods, Results, Conclusions, Figures and Tables with Captions, Funding Information, Author Contributions, Conflict of Interest and other Ethics Statements. Check the Journal Instructions for Authors for more details.“ I believe this means that the structure of the article is the usual one. Please check other papers published there.

3.      It is claimed in the abstract that none of the used waste materials chemically reacted with the clay during firing. How did you prove that?

4.      Bigot and Pffeferkorn method are not well defined again, since it is not clear from the text ion which way were the numbers obtained/calculated.

5.      You determined thermal conductivity coefficient by using EN1745 standard, which is actually a set of equations to calculate this coefficient based on the thickness of the walls. How did you measure the thickness on laboratory samples that had no voids?

6.      Please, delete the term “logical” in line 134.

Reviewer 2 Report

The manuscript is improved and changed according to the comments. However from my point of view, i still think that the materials and methods part is more appropriate to put before the results.
